# CONFIDENCE-GUIDED MCTS FOR EFFICIENT LONG-HORIZON WEB AGENT TASKS

## ABSTRACT

LLM agents that solve long-horizon tasks on the web often rely on Monte Carlo Tree Search (MCTS) to plan and reason over extended trajectories. While effective, standard MCTS requires wide branching and repeated value evaluations, making it very compute-intensive. We introduce confidence-guided MCTS, a method that uses internal certainty signals from the model's own log-probabilities to efficiently allocate search power to MCTS. The guided MCTS enables adaptive branching that adjusts the width of the tree depending on how confident the model is, reducing expansion when predictions are already decisive and vice versa. Our paper also includes multiple variants for integrating confidence into tree search; variants like weighted backpropagation incorporate certainty directly into value updates, amplifying reliable rollouts and reducing the impact of noisy ones. The method demonstrates that lightweight internal signals can guide search more effectively, reducing inference computation while preserving or even improving success on complex long-horizon tasks, moving closer to the Pareto frontier. Confidence-guided MCTS highlights a simple but powerful direction: using the model's own certainty to make search-augmented agents more efficient without extra supervision.

## 1 INTRODUCTION

Large Language Model (LLM) agents have rapidly advanced in their ability to plan, reason, and interact with complex environments (Wang et al., 2024). However, solving long-horizon tasks, those requiring many steps of reasoning or interaction, remains a persistent challenge (OpenAI, 2025; Anthropic, 2025; Comanici et al., 2025). Recent multimodal benchmarks such as VisualWebArena (Koh et al., 2024) have exposed the limitations of current approaches, as agents must not only parse multimodal instructions but also perform lengthy web interactions with high precision. Standard prompting approaches (Yao et al., 2023) often struggle to maintain coherence and efficiency over extended trajectories, highlighting the need for structured search methods.

Monte Carlo Tree Search (MCTS) provides a principled way to balance exploration and exploitation in sequential decision-making (Silver et al., 2017). When integrated with LLMs, MCTS enables the agent to simulate multiple possible futures and choose promising ones. For instance, Yu et al. (2025) propose ExACT, combining reflective MCTS with exploratory learning to train LLM-based agents to refine their strategies. Similarly, RoT (Hui & Tu, 2024) guides weaker models by distilling tree-based reflections from stronger models. While these methods achieve significant improvements, their computational demands are high: large search trees, long rollouts, and repeated value evaluations can cause prohibitive token usage and latency.

Recent works have turned toward making search more efficient. Chen et al. (2025) introduce Boosting of Thoughts, which refines reasoning through iterative exploration and error correction, yielding more targeted expansions of the search tree. Yang et al. (2025) show that enabling models to backtrack during inference reduces wasted exploration and mitigates overthinking. EquivPruner (Liu et al., 2025) further improves efficiency by pruning redundant branches that are semantically equivalent, reallocating computation via adaptive branching and exploration without changing the value-call. Despite these innovations, a key question remains: can we leverage the LLM's own internal signals to decide when and how to expand the search tree, rather than relying solely on external rollouts or handcrafted heuristics?

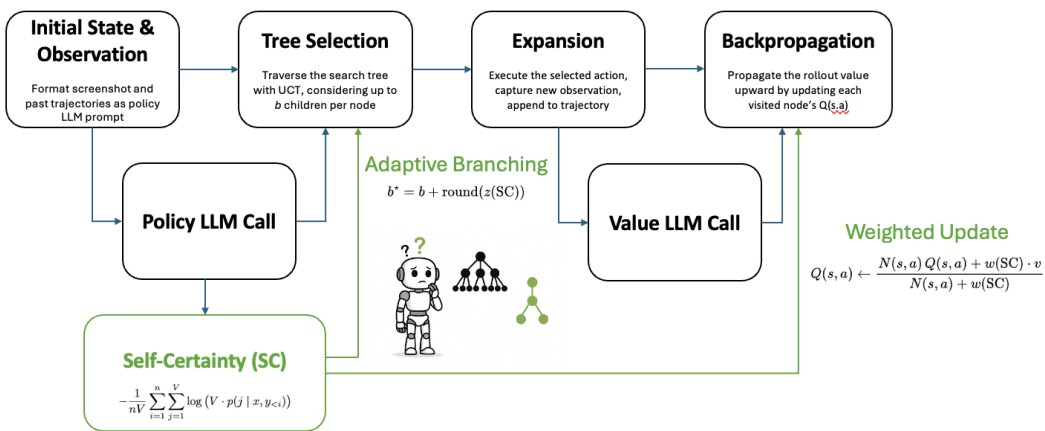

Figure 1: Diagram of self-certainty–guided adaptive branching and weighted updates in MCTS. Here, SC denotes self-certainty, $z$ its standardized $z$-score relative to historical self-certainty. The function $w(\cdot)$ nonlinearly scales the confidence signal; details are provided in Section 3.

To address this challenge, we propose the confidence-guided MCTS method. We use self-certainty, computed from token-level logprobs, as a lightweight internal signal that tracks how decisive the model is. Treating this signal as a proxy for confidence, our search adaptively adjusts tree parameters.

In particular, self-certainty enables adaptive branching, which allocates more expansion effort only when the model is less confident. Self-certainty weighted backpropagation stabilizes value estimates by amplifying decisive rollouts and dampening noisy ones, guiding the tree toward stronger actions with fewer evaluations. On VisualWebArena, both methods reduced token usage while keeping success above to baseline, with backpropagation often yielding smoother convergence and adaptive branching offering the largest savings on complex trajectories. Together, these methods show that internal confidence can serve as a lightweight control signal, making long-horizon agents more efficient.

Experiments on VisualWebArena show that this approach consistently reduces token usage while maintaining success rates comparable to the baseline. Variants such as adaptive branching and confidence-weighted backpropagation achieve notable efficiency gains, whereas unstructured exploration strategies consume more tokens without clear benefits. These results highlight that internal confidence signals can enable search-augmented agents to better allocate compute and solve long-horizon tasks more efficiently.

## 2 RELATED WORK

**Search-augmented reasoning.** MCTS has been central to recent attempts at scaling reasoning in LLM agents (Xie et al., 2024; Zhang et al., 2025; Antoniades et al., 2024). Yu et al. (2025) use reflective rollouts to provide explicit feedback during tree search, enabling agents to learn exploratory strategies that transfer across tasks. RoT (Hui & Tu, 2024) enhances this by summarizing prior search trajectories into reusable guidelines, effectively compressing the value of deep searches into lightweight heuristics. Beyond these, more dynamic strategies have emerged. Boosting of Thoughts (Chen et al., 2025) employs trial-and-error prompting to iteratively refine candidate reasoning paths, while Self-Backtracking (Yang et al., 2025) explicitly trains models to recognize when to abandon unproductive lines of thought. EquivPruner(Liu et al., 2025) complements these by pruning equivalent or redundant expansions, improving both efficiency and accuracy. Together, these approaches reflect a shift toward adaptive and resource-aware search in long-horizon reasoning.

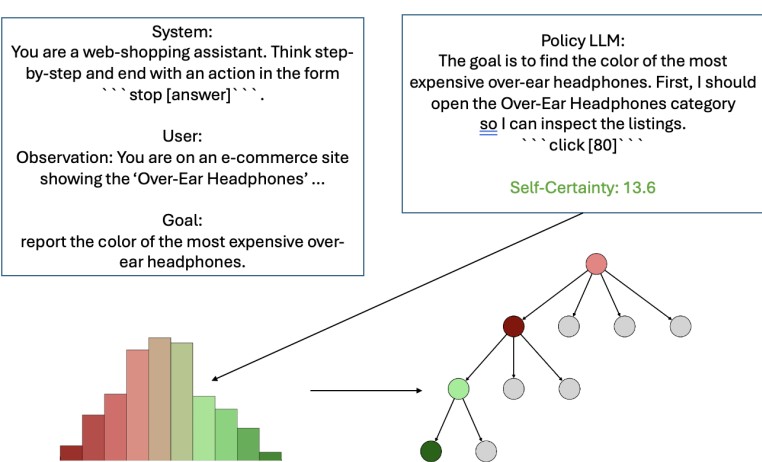

Figure 2: Given a self-certainty signal from the Policy Call, the Adaptive Branching method adjusts the expansion width according to the relative position of the self-certainty within the distribution, as reflected by the z-score.

**Intrinsic signals and internal feedback.** While external evaluators or human feedback can guide LLMs, recent work demonstrates that intrinsic signals can serve as reliable proxies for correctness (Prabhudesai et al., 2025). Kang et al. (2025) show that self-certainty enables scalable best-of-$N$ selection by distinguishing correct outputs without reference answers. Zhao et al. (2025) extend this idea through Intuitor, using self-certainty as the sole reward in reinforcement learning from internal feedback, matching or exceeding the performance of reward-based methods. Deep Think with Confidence (DeepConf) (Fu et al., 2025) further demonstrates that confidence signals can filter low-quality reasoning traces, yielding substantial token savings. These studies suggest that internal confidence measures can function as lightweight yet powerful tools for improving inference. Our work builds on this trajectory by using self-certainty directly into the control loop of MCTS, providing a mechanism for adaptive allocation of compute during search.

## 3 METHOD

To improve the efficiency of Monte Carlo Tree Search (MCTS) on long-horizon web tasks, we leverage self-certainty as an internal confidence signal derived directly from the language model's log-probabilities. Instead of expanding or simulating every branch uniformly, the tree considers how decisive the policy distribution already is before allocating additional compute. Given a partial trajectory $(x, y_{<i})$, self-certainty is defined as

$$
\text{SC}(x, y_{<i}) = -\frac{1}{nV} \sum_{i=1}^{n} \sum_{j=1}^{V} \log \left( V \cdot p(j \mid x, y_{<i}) \right),
$$

where $n$ is the generated length, $V$ is the effective vocabulary size, and $p(j \mid x, y_{<i})$ is the model probability assigned to token $j$. Low self-certainty indicates diffuse distributions, while high self-certainty reflects more decisive predictions. In practice, we only observe a partial vocabulary from the API (top-k). We therefore compute self-certainty over an effective support consisting of the top-k tokens, and replace V with the effective size $V^\star = k$.

In our setting, the model prompt $x$ is constructed from the current environment state $s$ (e.g., the observation/DOM snapshot plus the system instruction), i.e., $x$ is a function of $s$. We therefore treat self-certainty as state-conditioned, writing $\text{SC}(s)$ for the self-certainty at state $s$. For readability, when the state is clear from context, we drop the argument and simply write SC.

We propose *adaptive branching* (see diagram 2) as the most direct way to incorporate self-certainty into MCTS: the branching factor expands when the model is uncertain and contracts when it is

confident. To test the generality of the signal, we also introduce it at other stages: in exploration, and in the value model's prompt, and in backpropagation. These variants help assess whether self-certainty improves search broadly, or only when shaping branching.

## 3.1 ADAPTIVE BRANCHING

We adjust the branching budget using standardized self-certainty. Let $b$ denote the base branching factor and $z(\text{SC})$ the $z$-score of self-certainty computed relative to statistics collected on a held-out set. The adaptive branching factor is

$$b^\star = b + \text{round}(z(\text{SC}))$$

Where round rounds to the nearest integer. formulation expands more broadly when the model's predictions are unusually uncertain and narrows expansion when predictions are unusually confident. Unless otherwise specified, we use a base branching factor of $b = 5$. The $z$-score statistics $(\mu, \sigma)$ are computed from baseline runs under the same model and decoding settings on a held-out subset of tasks in the same suite.

## 3.2 GUIDED EXPLORATION

PUCT is a widely used version of Upper Confidence bounds(UCT) that incorporates a policy prior $P(s, a)$ into the computation of MCTS selection score popularized by Silver et al. (2017). For our method, we adjust the the exploration bonus in PUCT through the self-certainty signal. For a state $s$ and action $a$, the PUCT score's second term is scaled adaptively by self-certainty instead of a exploration constant, which becomes:

$$U(s, a) = Q(s, a) + w(\text{SC})\, P(s, a) \frac{\sqrt{N(s)}}{1 + N(s, a)},$$

where $P(s, a)$ is the policy prior, $Q(s, a)$ the empirical value, and $N(\cdot)$ the visit counts. The coefficient $w(\text{SC}) = \exp(-z(\text{SC}))$. The negative sign aligns with the intent to decrease exploration when self-certainty is high and increases it when self-certainty is low. This ensures that uncertain nodes receive stronger exploration pressure while confident nodes are explored more conservatively.

## 3.3 AUGMENTED PROMPT

Simulations still use the same value LLM call, but we add a single summary line describing the most recent confidence estimates (practically the last four self-certainty and z-score pairs). This gives the evaluator a view of how decisive the policy has been, so value judgments reflect both the trajectory and the agent's internal certainty, while adding only a handful of tokens. Ideally this should be performing similar to next method.

## 3.4 WEIGHTED BACKPROPAGATION

Backpropagation updates value estimates by weighting them with self-certainty. If a node $s$ receives rollout value $v$ with associated self-certainty SC, we update

$$Q(s, a) \leftarrow \frac{N(s, a)\, Q(s, a) + w(\text{SC}) \cdot v}{N(s, a) + w(\text{SC})}$$

We keep visit counts as integers $(N(s, a) \leftarrow N(s, a) + 1)$ and define the confidence weight as $w(\text{SC}) = \exp(z(\text{SC}))$. Here, $w(\text{SC})$ serves as an importance weight: trajectories judged more confident contribute more strongly to the running value estimate, while uncertain ones have reduced impact. This way, confidence is not merely tracked but actively modulates the signal being propagated upward in the tree.

In conclusion, self-certainty dynamically reshapes the search by adjusting branching, guiding exploration, and informing simulation and backpropagation. This leads to a more efficient allocation of compute, focusing effort where the model itself is uncertain.

| Metric | Corr. with Value | Corr. with Success |
|---|---|---|
| Self-certainty | 0.33 | 0.17 |
| Negative Entropy | 0.32 | 0.16 |
| Negative Perplexity | 0.30 | 0.15 |
| Negative Gini impurity | 0.31 | 0.16 |
| Top1–Top2 diff | 0.31 | 0.18 |

Table 1: Spearman correlation of mean logprob-derived metrics with mean rollout value and task success. Entropy, perplexity, and gini impurity were logged in negative form, so their correlations are shown here as positive for interpretability.

## 4 EXPERIMENTS

We evaluate confidence-guided MCTS on VisualWebArena (Koh et al., 2024), which provides multimodal, long-horizon web tasks across Reddit-like discussions, classifieds, and e-commerce shopping suites. The goal is to determine whether self-certainty aware search can reduce token usage while maintaining task success compared to the standard MCTS configuration.

### 4.1 SELF-CERTAINTY CORRELATION

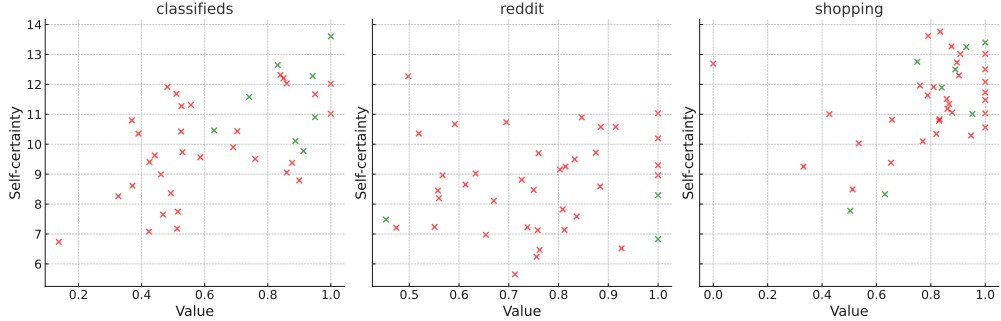

Figure 3: Scatter plot of mean value vs. mean self-certainty, colored by task success (red: fail, green: success). On classified and shopping tasks, we see clear correlation between self-certainty and value, with most successful run also aggregating in high self-certainty and high-value tasks.

Before directly testing our methods, we want to investigate whether self-certainty and other logprobs-based metric can be a good predictor of task success or at least give signal on whether a trace is worth exploring, we plot the relationship between self-certainty and rollout value judged by LLM in Figure 3. The scatter shows that successful task runs largely aggregate in the high-confidence region, which also corresponds to higher value scores. This suggests that self-certainty provides meaningful signal for distinguishing promising trajectories.

**Metric definitions.** Let $p(j \mid x, y_{<i})$ denote the next-token distribution over a vocabulary of size $V$ at position $i$, and write $p \in \mathbb{R}^V$ for brevity, with sorted values $p_{(1)} \geq p_{(2)} \geq \cdots \geq p_{(V)}$. When aggregating across a sequence of length $n$, we normalize each metric by $\frac{1}{n}$ to reduce length bias:

$$\textbf{Entropy:} \ H(p) = -\frac{1}{n}\sum_{i=1}^{n}\sum_{j=1}^{V} p_j \log p_j \qquad \textbf{Perplexity:} \ e^{H(p)}$$

$$\textbf{Top1–Top2 margin:} \ \frac{1}{n}\sum_{i=1}^{n}\big(p_{(1)} - p_{(2)}\big) \qquad \textbf{Gini impurity:} \ \frac{1}{n}\sum_{i=1}^{n}\Big(1 - \sum_{j=1}^{V} p_j^2\Big)$$

**Analysis.** Figure 3 and Table 1 show a consistent, moderate relationship between confidence and rollout quality. Across metrics, correlation with value is in the low to mid 0.3 range, and correlation

| Method | Classifieds | | | Reddit | | | Shopping | | |
|---|---|---|---|---|---|---|---|---|---|
| | Success | Prompt | Completion | Success | Prompt | Completion | Success | Prompt | Completion |
| ReACT | 0.100 | 0.26x | 0.02x | 0.050 | 0.33x | 0.02x | 0.075 | 0.19x | 0.04x |
| MCTS | 0.250 | 1.00x | 1.00x | 0.100 | 1.00x | 1.00x | 0.175 | 1.00x | 1.00x |
| Branching | **0.250** | **0.84x** | 0.83x | **0.100** | 0.95x | 0.97x | 0.200 | 1.03x | 0.99x |
| PUCT | 0.225 | 1.10x | 1.15x | 0.075 | 1.17x | 1.20x | 0.200 | 1.29x | 1.23x |
| Prompt | 0.175 | 1.09x | 1.08x | **0.100** | 1.03x | 1.05x | 0.225 | 1.03x | 0.99x |
| Backprop | **0.250** | 0.88x | **0.94x** | **0.100** | **0.92x** | **0.91x** | **0.250** | **0.91x** | **0.89x** |

Table 2: Performance of different methods on VisualWebArena, 40 tasks per suite. Token usage is reported as multiples of the suite-specific baseline. Bold entries are the best result among self-certainty variants.

with task success is weaker but still positive. The effect is clearest in Shopping, where successful runs cluster in the high-value, high-certainty region. Reddit shows the weakest separation. These patterns suggests that self-certainty is a useful routing signal for compute, though not strong enough to act as a decision rule on its own.

## 4.2 MAIN EXPERIMENT

All methods use `gpt-4o-mini` for policy generation, self-certainty queries, and value estimation. Policy and self-certainty calls use temperature $1.0$ with top-$p = 0.95$, and the API returns top-$k = 20$ logprobs. For self-certainty extraction, generations are capped at length of 100 to ensure stable statistics and reduce length effects.

We fix MCTS budgets across methods unless otherwise noted: branching factor $b = 5$, maximum depth $L = 4$, value-function budget is set as 20, time budget of 5 minutes per task, PUCT constant $c_{\text{puct}} = 1.0$, and at most 5 environment steps. Only the component under test (branching, exploration, prompt, or backpropagation) is varied. In experimental setting, all learning or reflection functionality are disabled on all variants to avoid length- and memory-induced shifts in self-certainty.

Evaluation is on 40 tasks per suite (Classifieds, Reddit, Shopping) using the official VisualWebArena success metric. Token usage is provider-reported and tracked separately for policy and value calls. Per-suite normalization statistics $(\mu, \sigma)$ for self-certainty are computed from separate baseline runs with the same model and decoding settings. We use the default benchmark environment, action set, viewport height, and UI/timeouts in ExACT framework. All other settings are held constant.

The baseline are ReACT and MCTS from ExACT framework without reflection to control for completion length and database effects on self-certainty. Previous method represented by ReACT is also used as baseline comparison. We further report four ablations that switch on exactly one confidence-guided component at a time: (i) Branching (SC-based adaptive branching factor), (ii) PUCT (SC-modulated exploration), (iii) Prompt (SC-augmented value prompt), and (iv) Backprop (SC-weighted value updates). Baseline MCTS uses none of these components.

**Analysis.** Table 2 indicates that confidence guidance mainly improves efficiency without reducing success, and in some cases improves success. The pattern is consistent across suites, with the largest gains on Shopping. As a weaker baseline, ReACT uses very few tokens yet the performance is also much worse than all methods in comparison, showing there is normally a compute performance trade-off.

Figure 3 helps explain this difference: the correlation between self-certainty and value is stronger on Shopping, where successful runs cluster in the high-certainty, high-value region. In contrast, Classifieds shows a weaker but still positive trend, while Reddit exhibits the weakest separation, making confidence less predictive and limiting the gains.

*Branching.* Confidence-aware branching reduces token usage relative to MCTS while keeping success unchanged. The controller narrows width when confidence is high and widens only under uncertainty, trimming many potentially low-value expansions. On Shopping, where confidence aligns closely with rollout value, this leads to large savings. On Reddit, where confidence-value correlation is weaker, branching still saves tokens but does not improve success.

*Backpropagation.* Confidence-weighted backpropagation offers the best overall balance. In Classifieds and Reddit it matches baseline success while using fewer tokens, and in Shopping it improves both efficiency and success. The mechanism is straightforward: confident rollouts contribute more to $Q$, stabilizing value estimates and preventing the search from revisiting weak branches. The example 4.2 illustrates this effect clearly. The backprop variant reinforces a decisive path through category to item to color, converging on the correct stop (*Black*) with high confidence, while the baseline explores side branches, fails to converge, and misses the answer. Token usage reflects this difference: backprop consumes less than half the prompt tokens of baseline, showing how confidence-weighted updates concentrate compute on the strongest trajectory.

```
Example: Backprop vs Baseline MCTS

Intent: color of the most expensive over-ear headphones

#click [80] (Over-Ear Headphones)
backprop: local z = +0.7 # positive z steers backup to reinforce this path
baseline: continues default exploration # keeps evaluating side branches

#click [60] (Sony WH-1000XM4)
backprop: Q=1.00, z = +1.0 # strong positive z, Q keeps climbing on same path
baseline: Q oscillates since returns are unweighted    # more side exploration

#stop [Black]
backprop: Q=0.86, z = +0.7 # confident stop, success (score 1.0)
baseline: includes "Black" # but no decisive signal, failure (score 0.0)

tokens:
backprop: 2.0M prompt
baseline: 5.7M prompt
```

*PUCT.* Self-certainty modulated exploration increases search breadth and raises token consumption. This occasionally helps, particularly in Shopping where broader lookahead is useful, but overall it is less compute efficient than backpropagation and offers limited benefit in Reddit or Classifieds.

*Prompt augmentation.* Adding confidence summaries to the value prompt introduces small overhead with mixed effects on success. Compared to branching and backpropagation which inject efficient confidence signals, prompt-level augmentation provides marginal additional benefit.

### 4.3 VARIANTS OF CONFIDENCE METRIC

We compare different formulations of self-certainty, including entropy, perplexity, the difference between top-1 and top-2 probabilities, and Gini impurity. These variants allow us to test whether other logprob–based measures can match or exceed the effectiveness of our proposed self-certainty. The main motivation that we pick self-certainty is based on its performance in (Kang et al., 2025) and higher correlation with value function in section 4.3.

**Analysis.** Table 3 swaps the same adaptive-branching controller across several logprob-based signals, so most differences come from how each metric behaves after standardization. Entropy keeps most nodes near the mid-range once we z-score it, so $b^\star$ only become unstable when the distribution really flattens, which is a good fit for the heuristic and the only variant that nudges success up-

| Method | Success | Prompt | Completion |
|---|---|---|---|
| Self-certainty | 0.100 | 1.00x | 1.00x |
| Entropy | 0.125 | 1.01x | 0.98x |
| Gini | 0.100 | 0.98x | 0.97x |
| Perplexity | 0.100 | 1.15x | 1.14x |
| Top1–Top2 diff | 0.100 | 1.00x | 0.97x |

Table 3: Comparison of self-certainty metrics on the reddit suite. Token usage is reported as multiples of the adaptive branching baseline.

ward. Perplexity is simply exponentiation of entropy. This stretches the tail and makes high confidence cases look more uncertain after z-scoring, so the controller is less sensitive than entropy. Gini impurity flattens distinctions once mass is already spread out, so its z-scores cluster close to zero; the branching rule then behaves almost identically to the self-certainty baseline. The top1–top2 gap

ignores how the remaining mass rearranges, so its z-scores oscillate whenever the second highest probability token changes, and producing behavior nearly indistinguishable from the baseline. In short, all metrics share the same pipeline; entropy happens to give the cleanest signal on this suite, while the others either over-amplify or dampen the same information.

## 4.4 GROUP BASED AGGREGATION

Inspired by recent work such as DeepConf (Fu et al., 2025), which shows that local confidence signals can better capture reasoning quality than global averages, we investigate aggregation methods to reduce length bias in self-certainty. Rather than averaging over all tokens, we study group-based aggregation.

Let a completion of length $L$ tokens be partitioned into $n$ groups, each of size $|G_i|$, so that

$$n = \frac{L}{|G_i|} \qquad SC_{G_i} = \frac{1}{|G_i|} \sum_{t \in G_i} C_t, \quad 1 \leq i \leq n$$

We focus on two variants of group aggregation in this experiment:

$$SC_{\min} = \min_{1 \leq i \leq n} SC_{G_i} \quad \text{(lowest-confidence group)} \qquad SC_{\text{tail}} = SC_{G_n} \quad \text{(tail group)}$$

Considering that the average policy completion length is approximately 80 tokens, we set the group size $|G_i| = 10$. These group methods aim to utilize the regions that will most likely determine the final action or reveal uncertainty, reducing the noise from less informative groups. By focusing on these critical groups, we test whether chosen local signals carry stronger predictive power than our baseline of normalized averages under adaptive branching.

| Method | Reddit | | | Shopping | | |
|---|---|---|---|---|---|---|
| | Success | Prompt | Completion | Success | Prompt | Completion |
| Sequence Average | 0.100 | 1.00x | 1.00x | 0.200 | 1.00x | 1.00x |
| Lowest-Conf Group | 0.100 | 1.05x | 1.10x | 0.200 | 0.82x | 0.87x |
| Tail Group | 0.100 | 0.98x | 0.92x | 0.175 | 1.08x | 1.07x |

Table 4: Performance of different grouping methods on 40 Reddit and Shopping tasks. Token usage is reported as multiples of default adaptive branching MCTS.

**Analysis.** As shown in Table 4, the results across Reddit and Shopping are mixed, with no consistent advantage for one grouping strategy. Compared to DeepConf (Fu et al., 2025), where group-based confidence signals yield clear gains on long-chain reasoning tasks, our policy model calls appear more stable due to their shorter completion length. In this setting, both lowest-confidence and tail groups occasionally reduce token usage but do not consistently improve success rates over the sequence average baseline. A qualitative inspection suggests that the tail group, which typically contains the action decision, tends to have more extreme self-certainty and has the potential to save tokens on some tasks; however, missed task on shopping arise from borderline self-certainty scores that are rounded into the wrong category. We suspect that under shorter-horizon settings, group aggregation will not provide stronger signal, and its benefits may only become more pronounced in longer trajectories.

## 4.5 ABLATION

We replace $z(\text{SC})$ with $r \sim \mathcal{N}(0, 1)$ in the branching rule

$$b^\star = b + \text{round}(r),$$

while keeping all other components unchanged. This serves as a control to check whether improvements stem from meaningful confidence signals rather than random noise.

| Method | Reddit | | | Shopping | | |
|---|---|---|---|---|---|---|
| | Success | Prompt | Completion | Success | Prompt | Completion |
| MCTS | 0.100 | 1.00x | 1.00x | 0.175 | 1.00x | 1.00x |
| Branching | 0.100 | 0.95x | 0.97x | 0.200 | 1.03x | 0.99x |
| Randomized | 0.075 | 0.94x | 0.95x | 0.175 | 1.03x | 0.97x |

Table 5: Performance of baseline, branching, and randomized methods on Reddit and Shopping. Token usage is reported as multiples of the MCTS baseline. On these tasks, the token usage differences are minor.

**Analysis.** Replacing self-certainty with random Gaussian noise in the branching rule reduces stability and hurts outcomes without yielding token savings. Success rates drop modestly on both Reddit and Shopping, while prompt and completion usage stay nearly unchanged (Table 5). This happens because random spikes in $b^\star$ add extra children with non-zero priors, diverting visits away from the strongest branch and lowering its $Q$ estimate, even though the overall number of rollouts and tree budget remain fixed. The example in appendix A.3 illustrates this clearly: the baseline holds $b = 5$, keeps the high-value branch prominent, and converges to the correct stop. While randomized branching expands to $b = 6$, spreads visits thinner, and ended at the wrong stop. Token usage stays similar across conditions because both strategies expand the tree to fill the same rollout and time budgets, the difference is how effectively that budget is allocated.

In conclusion, self-certainty provides a stable signal to concentrate compute, while randomized branching wastes capacity on side branches and degrades success at the same cost. From the quantitative results, randomized branching does not yield improvements over baseline MCTS and can even worsen the compute–performance trade-off on these tasks, reinforcing that self-certainty provides genuine merit as a guiding signal.

## 5 DISCUSSION

Our experiments show that self-certainty is broadly effective for guiding MCTS. Confidence-weighted backpropagation is particularly strong, simultaneously improving performance and reducing compute, while adaptive branching reliably decreases token usage with no reduction in performance.

Different confidence formulations (e.g., entropy, perplexity, Gini impurity, top-1 vs. top-2 margin) do not change results much, as they are highly correlated. The main distinctions lie in how each re-shapes the distribution. For instance, perplexity amplifies variance relative to entropy. This suggests the value of introducing a tunable hyperparameter that controls signal strength and can be adjusted by heuristic. Group-based aggregation methods also show limited benefits under our setting, indicating that averaging suffices for short completions. Overall, the benefit of self-certainty is shown across methods and further supported by randomized ablations and correlation studies.

Future work should address current limitations and explore broader directions. Our evaluation is constrained by compute, preventing exhaustive runs across all tasks, and results may still reflect variance from multiple sources. Scaling to larger environments and more diverse tasks would provide stronger evidence. Additionally, longer-horizon domains may benefit from trajectory- or other form of window-based aggregation. Finally, internal signals beyond logprobs, such as probing internal activations represent promising alternatives for guiding search.

## 6 CONCLUSION

This work introduces self-certainty as a lightweight internal signal to guide MCTS in LLM agents. By integrating confidence into backpropagation and branching, we achieve more favorable compute–performance trade-offs without modifying the base model. Our analysis shows that the advantage of self-certainty is robust across variants, while randomized signals do not provide such benefits, suggesting it captures a general property of model behavior rather than incidental behaviors. The consistency of results points to self-certainty as a promising direction for scalable reasoning.

ETHICS STATEMENT

All experiments are conducted on publicly available benchmark tasks without the use of personal data or human subjects. Model queries were issued under standard API usage policies and within provider rate limits. Our goal is to advance methods for more efficient and reliable reasoning in LLM-based systems, not to promote misuse. By highlighting both the benefits and limitations of self-certainty signals, we aim to inform future research on safe and transparent integration of search strategies in agentic systems.

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

# A   APPENDIX

## A.1   USE OF LLM.

LLMs were used for text-related tasks such as revising, editing, and refining the clarity and readability of the paper.

## A.2   EXPERIMENT SETUP.

**Environments.** All agents interact with the VisualWebArena (VWA) benchmark via the official SOM interface ($1280\times2048$ viewport, "image_som" observations, 5 step budget, 2.5 s dwell between executions). VWA is hosted on an AWS EC2 g5.4xlarge (single NVIDIA A10G, CUDA driver 12.2) that also runs the BLIP2 captioning model (Salesforce/blip2-flan-t5-xl) required by the environment. All policy/value calls (details of calls can be found in appendix A.3) use the OpenAI API with the "gpt-4o-mini" chat model unless otherwise specified.

**Settings** The ReAct prompt baseline uses the MCoTPolicyPConstructor with temperature $1.0$, top-$p = 0.95$. The value model mirrors the policy model ("gpt-4o-mini"), while rubric re-scoring requests use temperature $0.7$, top-$p = 0.9$. Embedding lookups rely on "text-embedding-3-small". Execution parameters inherit the default RMCTS search scaffold (branching factor 5, depth limit 4, PUCT constant 1.0, value-function budget 20, 5-minute wall-clock cap).

All MCTS variants share the ReinforcedPolicyPConstructor backbone with self-certainty feedback. The policy head again uses temperature $1.0$, top-$p = 0.95$. The value head matches those limits but is called under the "vf" budget of 20 simulations per decision. No reflective rollouts are allowed (max reflections 0 for both policy and value) to use MCTS without reflection. Search is value-function guided (vf) with PUCT $= 1.0$ and the same 5-minute task timeout. All other hyperparameters (temperatures, embedding backend, parsing/repetition failure thresholds of 3/5) remain identical to the baseline.

A.3  EXAMPLE.

### Example: Policy Call

```
[
  {
    "role": "system",
    "content": "You are an autonomous web-browsing agent. You can issue actions such as
    'click [id]', 'type [id] [text] [1/0]', 'scroll [direction]', and 'stop [answer]'.
    Think step by step, refer to the observation, and output exactly one valid action."
  },
  {
    "role": "user",
    "content": [
      {
        "type": "text",
        "text": "Task: price of the most expensive feather lamp in \"Lamps & Shades\"\n
Current URL: http://ec2-54-157-58-23.compute-1.amazonaws.com:7770/\n
Open tabs: [0] One Stop Market\n
Previous action: type [7] [feather lamp] [1]\n
Observation:\n[13] [A] [Clothing, Shoes & Jewelry]\n[14] [A] [Home & Kitchen]\n...\n[106] [A] [Modern Luxury Ostrich
Reason step-by-step, then return your final line as 'Action: <command>'."
      },
      { "type": "text", "text": "IMAGES: (1) current page screenshot" },
      { "type": "image_url", "image_url": {"url": "data:image/png;base64,iVBORw0K..." } }
    ]
  }
]
```

### Example: Value Call

```
[
  {
    "role": "system",
    "content": "You are the value function for a web agent. Judge how close the current
    trajectory is to fulfilling the user's intent. Use the rubric, recent screenshots,
    and action history. Conclude with 'Thoughts: ...' and 'STATUS CODE: A/B/C/D/E'."
  },
  {
    "role": "user",
    "content": [
      {
        "type": "text",
        "text": "Intent: price of the most expensive feather lamp in \"Lamps & Shades\"\n
Rubric:\n1. Is the selected product a feather lamp?\n2. Is it in the \"Lamps & Shades\" category?\n
3. Is its price the highest among feather lamps?\n4. Does the response state the price explicitly?\n
Recent actions:\n1. type [7] [feather lamp] [1]\n2. click [112]\nCandidate stop action: stop [$49.99]\n\n
Evaluate whether the agent is on track."
      },
      { "type": "text", "text": "IMAGES: (1) last page screenshot" },
      { "type": "image_url", "image_url": {"url": "data:image/png;base64,iVBORw0K..." } }
    ]
  }
]
```

### Example: Branching vs Randomized

```
Intent: price of the most expensive feather lamp in "Lamps & Shades"

#type [7] (feather lamp)
baseline: b = 5 # stable branching from self-certainty signal
randomized: b = 6 # more side branches appear, extra children steal visits from best path

#Next Action
baseline: Q = 0.78 # fewer active children; high-value branch stays prominent
randomized: Q = 0.56 # exploration diverted to new branch

#Result
baseline: stopped at [$920.99] # Success (score 1.0)
randomized: stopped at [$788.88] # Failure (score 0.0)

tokens:
baseline: 0.98M prompt
randomized: 0.98M prompt
```

