# OpenReview forum: "Confidence-Guided MCTS for Efficient Long-Horizon Web Agent Tasks"
_ICLR.cc/2026/Conference — Submitted to ICLR 2026_

### Official Review · Reviewer_wC9h · 2025-10-17

**Soundness:** 1
**Presentation:** 1
**Contribution:** 2
**Rating:** 2
**Confidence:** 3

**Summary:**

This paper introduces new heuristic search (as well as backprop.) policy/s for MCTS, with the intent of improving LLM performance in long horizon tasks.

**Strengths:**

1. The idea of exploiting specific properties of LLMs to direct the search of MCTS seems promising.
2. The paper evaluates and compares multiple possible metrics of certainty for directing MCTS for LLMs.

**Weaknesses:**

1. Theoretical analysis: Specifically, an analysis of the consequences of changing the backprop. and search of MCTS with this hueristic is missing. What policy does MCTS approximate with these changes?
2. Theoretical motivation is missing. Why use certainty in this way and not in other ways? For example, as another value / reward term to search with respect to, rather than a factor on the policy-prior?
2. Results analysis: The paper focuses on self-certainty, a (very, 2025) recently introduced metric for certainty in LLMs. Despite the results apparantly showing that other metrics of certainty perform better.
3. Presentation:
    1. Key concepts are never introduced or explained which makes the paper and the choices of notation hard to follow. In more detail below.
    2. Key citations are missing, or follow much later than when they are first referred, making certain components (self-certainty) seem like contributions of the paper - where they are not.
    3. Results presentation in unclear. In more detail below.
4. Statistical significance: not measured or discussed.

**Questions:**

Key concepts are never introduced or explained:
1. MCTS is never introduced or explained.
2. How MCTS is used with LLMs is not introduced or explained, which makes the paper hard to follow. Introducing notation for 1 and 2 will also make the paper easier to discuss.
3. The idea of branching / a branching factor is never introduced, explained or defined. Im not familiar with a branching factor being standard in MCTS: the search process is defined by the action space $A$ and the prior policy $\pi_\theta$ in PUCT.
4. The baseline REACT is never introduced or explained. Is it MCTS-based, search-based, other?
5. The definition / description of z(SC) is unclear.
6. The equation of PUCT (which is unnumbered and would be easier to refer to had it been numbered) uses P(s,a) to refer to the policy prior. However, isn't the LLM's logits for which token to output $p(j | x, y_{<i})$ exactly this policy prior? If that is the case, it seems confusing and unnecessary to denote the same thing twice in two different ways. Also, what state and actions are in this setting (i.e. point 2) is never defined, which makes it further unclear how this should be read in this context.

Key citations are missing:
1. The citation of MCTS cites AlphaZero, not MCTS. Since MCTS is perhaps the most important mechanism in this work, I would cite it.
2. PUCT and UCT are not cited.
3. Self-certainty is first introduced in Figure 1 and intro, but only referenced later in the related work, leaving a possible impression that this is one of the contributions of the authors.
4. REACT is not cited.

Results presentation is unclear:
1. In Table 2 baselines and ablations are presented, but is a complete agent using all the changes evaluated? The MCTS row seems to be a baseline, not a combined agent.
2. In Table 2 bolds are presented, but it's not clear what they denote. There is no metric for stat. signif., so these are not the stat. signif. best results. It's also not clear if these are the best results / take the baselines-row into account.
3. What performance metrics exactly are used / how the results in the table should be read (higher better / lower better?) is unclear.

Additional comments:
1. Self certainty should be introduced, defined and cited in a background section, not method section.
2. In MCTS / RL V is traditionally used for value. Using another letter to denote vocabulary size (and explaining where the values used in MCTS with LLMs are coming from) will help reduce confusion for RL readers.
3. $p(j | x, y_{<i})$ is defined twice, and denoted in two different ways (the above and $p_j$) which is confusing.

---

### Official Review · Reviewer_vEZs · 2025-10-31

**Soundness:** 3
**Presentation:** 3
**Contribution:** 2
**Rating:** 2
**Confidence:** 4

**Summary:**

The paper proposes Confidence-guided MCTS to allow LLMs to utilize MCTS for solving long-horizon tasks efficiently while retaining the same performance as MCTS. The key insight is that the LLMs own certainty as an internal signal as a heuristic during the MCTS process.

The authors using this heuristic in different ways. Adaptive branching dynamically increases/decreases the branching factor available to MCTS based on the models certainty. For high certainty, fewer options need to be considered and vice versa. The authors also explore using self-certainty in other parts of the MCTS process.

They showcase some intuitiion of their approach by measuring corelation and then perform an empirical evaluation with gpt-4o-min on visual web arena. They also conduct ablations that check different confidence metrics, group-aggregation and effect of random noise.

**Strengths:**

1. The paper is very clear and well-written

2. The idea is novel and intuitive in its applicaiton to MCTS.

3. The framework seems flexible to allow different confidence metrics and in different parts of MCTS.

**Weaknesses:**

I think this paper offers quite a bit of merit in the ideas but I think that the empirical design has made the score lower than it needs to be.

1. The paper only considers MCTS and ReAct as baselines. Why not consider other approaches like RoT, ToT etc. MCTS only has 25% success rates so if these other baselines have more, then MCTS as a whole might not be useful and you would need to probably show that your approach can be adapted to SOTA techniques for it to have practical merit.

2. The main claim is that the efficiency of MCTS is improved without impacting success rates. The writing in lines 310-332 is a bit poor and does not showcase the core strengths. For example, the first sentence says that confidence improves efficiency but it is not mentioned by how much. Looking at table 2, there seem to be < 10% improvement but these are relative numbers and it is very had to estimate if this 10% improvement is great.

There is data missing here since multipliers are used. Saving 1 token in 10 tokens is a 10% improvement but not impressive. Please provide this data.

3. Another big concern is that only gpt-4o-mini is considered. Why not some open-source modesls?

4. The analysis should also showcase or perhaps expand on a trace where fewer tokens are used with the same success. This is important since it is clear from Table 2 that confidence metrics are domain-dependent in their performance and can actually be worse than vanilla MCTS in certain settings (eg. Branching in Shopping-prompt with 1.29x/1.23x). How would one know which strategy to select. Here backprop seems to work, but its only one domain. Probably more domains might need to be tried to give a more wholistic view.

5. Ablation 4.5 does not make sense since r is sampled from [0, 1]. The z-score will have a different range i assume so the range sampled for noise should be the same as that of the z-score in adaptive branching. Ablation 4.5 is never going to even reduce the branching factor so those results are not really conclusive.

I think 4.4 and 4.5 could move to an appendix and have results with more baselines and models.

**Questions:**

Overall, I like this work. I think it is an interesting idea. Im happy to discuss and raise my score on this paper provided some of my concerns are adquately addressed. My primary reason for a 2 is perhaps W2,  not having results with OSS models and not having any justificaiton on why other search-tree methods are not run as baselines.

---

### Official Review · Reviewer_K3xZ · 2025-11-01

**Soundness:** 2
**Presentation:** 2
**Contribution:** 2
**Rating:** 2
**Confidence:** 3

**Summary:**

This paper tackles the challenge of long-horizon task planning on the web by augmenting Monte Carlo Tree Search (MCTS) with a notion of the model’s own confidence. The proposed Confidence-Guided MCTS uses a lightweight internal self-certainty signal (derived from token-level log probabilities) as a proxy for how decisive the language model’s predictions are. This confidence signal is integrated into multiple stages of the MCTS pipeline. In effect, the agent allocates computation adaptively – avoiding wide exploration when the next action seems clear and only investing in broader search when needed. Empirical results on the VisualWebArena benchmark demonstrate that confidence-guided search can reduce token usage significantly while preserving, or even slightly improving, task success rates relative to a standard MCTS baseline.

**Strengths:**

•Proposes a novel MCTS variant that leverages the model’s own confidence signals (no extra supervision) to guide search and improve efficiency in long-horizon web tasks.

•Demonstrates strong empirical performance, significantly reducing computation (token usage) while maintaining or improving success rates on complex web tasks.

**Weaknesses:**

•	Incremental Contribution: While the idea of using the model’s own log-probability-derived confidence in the search process is interesting, it could be seen as a relatively incremental advance rather than a fundamentally new algorithmic insight. The paper builds on a line of recent work that has explored internal confidence signals for filtering or selection in LLM reasoning (e.g., Kang et al., 2025; Fu et al., 2025). The contribution here is mainly to apply such signals to guide MCTS heuristics (branching and backup). This is a natural extension of prior ideas and might be viewed as a heuristic tweak to MCTS for efficiency. The novelty is thus moderate – it’s a new combination of existing concepts (MCTS + confidence metrics) rather than an entirely new approach to long-horizon planning.

•	Limited Evaluation Scope: The experimental evaluation, although carefully controlled, has some limitations in scope. First, the benchmark tasks are relatively few (40 tasks in each of three suites) and of bounded length – the environment was run with a maximum of 5 interaction steps per task. This is a somewhat limited definition of “long-horizon.”

•	Baselines and Comparisons: The baselines chosen include a reasonable naive approach and a standard MCTS. However, the paper does not compare against more advanced recent methods that also aim to improve long-horizon reasoning efficiency, which were discussed in the related work. Approaches like Boosting of Thoughts (Chen et al., 2025), Self-Backtracking (Yang et al., 2025), or EquivPruner (Liu et al., 2025) are designed to prune or adjust search in intelligent ways.

**Questions:**

•	Combined Effects: Should the work evaluate the full confidence-guided MCTS with all components enabled simultaneously?

•	Longer Horizons and Scalability: The experiments were constrained to at most 5 environment interaction steps. Should the work try to test this approach on tasks requiring a deeper search or a greater number of reasoning steps?

•	Integration with Other Techniques: Is it possible to combine confidence-guided search with other recent efficiency improvements for LLM planning (such as backtracking, or heuristic pruning of search space)?

---

### Meta-Review · Area_Chair_b2m9 · 2026-01-07

**Summary:**

This paper introduces Confidence-Guided MCTS, a framework aimed at improving the efficiency of language model agents in long-horizon web tasks. The method utilizes internal certainty signals, derived from the model’s own token-level log-probabilities, to adaptively allocate search resources. Specifically, it employs adaptive branching to adjust tree width based on prediction confidence and uses weighted backpropagation to incorporate certainty into value updates. The goal is to reduce inference compute while preserving task success rates, with evaluation performed on the VisualWebArena benchmark.

**Reviewer Concerns:**

The reviewers are unanimous in their recommendation for rejection. While the reviewers acknowledge that the motivation to improve MCTS efficiency using internal feedback is intuitive and potentially impactful, they conclude that the submission is not yet ready for publication. Primary weaknesses include the incremental nature of the contribution, as the work combines existing certainty metrics with standard search heuristics without introducing fundamental new insights. The empirical evaluation is criticized for its limited scope, focusing on very short interaction horizons and a single closed-source model (gpt-4o-mini) while failing to compare the method against more advanced search-augmented baselines.

Further concerns involve the lack of theoretical rigor. The reviewers point out that the paper does not analyze how the proposed heuristic changes affect the convergence properties of MCTS or the specific policy being approximated. There are also several presentation flaws, as the submission omitted core citations for MCTS and PUCT, and presented results without clear statistical significance or consistent formatting.

**Reviewer Scores:**

No rebuttal or revision has been submitted. It is unlikely that any reviewer would have changed their score.

---

### Decision · Program_Chairs · 2026-01-26

Reject